

# The University Campus as a Learning Environment: the role of a Campus-based Living Lab in a Blended Teaching and Learning Environment

**Steven L. Rogers** [1]**, Adam J. Jeffery** [2]**, Jamie K. Pringle** [1] **Antonia C. Law** [1]**, Alexandre Nobajas** [1]**,**
**Katie Szkornik** [1]**, Angela C. Turner** [1] **& Adam Moolna** [1] **& Luke Hobson** [1]

[1] School of Geography, Geology and the Environment, William Smith Building, Keele University, UK.

[2] Keele Institute for Innovation and Teaching Excellence, Claus Moser Building, Keele University, UK.

*Correspondence to*: Steven L. Rogers (s.l.rogers@keele.ac.uk)

**Abstract.** "Living Labs" provide stakeholders with an authentic and spontaneous environment in which innovations and technologies can be developed. This paper highlights the use of Living Labs as an educational teaching and learning environment. We give examples of practice currently used and present a conceptual framework for pedagogic design of activities and assessment in a Living Lab environment. The examples provided are based around current HE under/post-graduate taught assessment and activities. We suggest that Living Labs, particularly campus based Living Labs, are an excellent opportunity for education providers to provide experiences for students that are realistic, promote empowerment of students, and are spontaneous, promoting student inclusivity and sustainability. Living Labs can introduce opportunities for inter- and transdisciplinarity and cross-cultural working and can provide an excellent base for education for sustainability.

## 1 Introduction

Living Labs are a reasonably recent concept arguably first coined in the 2000s (Markopoulos and Rauterberg, 2000) and are traditionally set up and used as research environments. There are a number of definitions used in the literature which are similar, typically corresponding to the definition used by Hossain et al. (2019) "A living lab is a physical or virtual space in which to solve societal challenges, especially for urban areas, by bringing together various stakeholders for collaboration and collective ideation". Because Living Labs are reasonably recent (particularly those on university campuses) there aren't many examples of deviation from the research focus and concept of the founding examples that the authors are aware of. Some examples of the educational value of Living Labs have been explored (Callaghan and Marlien, 2015; Mazutti et al., 2020).

This paper aims to initially outline a number of teaching and learning activities that take place at Keele University outside of the traditional classroom, lecture theatre or laboratory and explain the educational framework around such activities. The activities have been used to provide authentic learning experiences to HE students, within the setting of a campus-based Living Lab. We then highlight how running such authentic assignments and assessments on campus can provide an





important element of a hybrid/blended learning environment and provide teachers and learners with an array of experiences and opportunities. These experiences can form important opportunities for students to widen their knowledge, experience and improve on skills which may have been covered digitally or in the classroom. We finally also introduce the concept of the Campus as a Classroom, making use of a campus-based Living Lab as a space for active learning in a curriculum

delivered via a range of modalities. Sustainable reflections on our field course travel (and associated $CO_2$ emissions) have factored into the justification using campus-based activities more, as aligned with the University's sustainability ethos.

As practitioners, the authors and colleagues have been making use of Keele campus-based activities for some years, however the COVID-19 pandemic accelerated the development and design of these activities and placed a renewed emphasis on the

use and benefits of campus as a Living Lab, and of the organisation of such activities. During the COVID-19 pandemic education providers globally have had to redesign and redevelop teaching provision. Whilst some changes have been sub-optimal alternatives to previous delivery, many innovations have provided platforms for efficient and effective learning environments. Another effect of the pandemic was educators having to reflect on what is/was available – one resource which was potentially underused and/or undervalued in many cases was the University campus – the reasons for this are not exactly

clear, but anecdotally can be linked to the effort needed to organise activities, the perceptions of using campus rather than travelling being seen an inferior choice, and/or educators being unaware of the possibilities of using campus space as an educational environment.

We hope the case studies presented here may provide inspiration for others interested in providing authentic, realistic,

spontaneous, immersive and empowering learning environments for their students. We also provide some thoughts on framing campus-based activities within a blended teaching environment, supporting and scaffolding authentic teaching and learning activities with asynchronous digital materials. The campus Living Lab can provide an accessible location for experiential learning and its use is potentially a much more sustainable alternative to many other field-based teaching. It is hoped that the reporting of the activities outlined here, and their framing within pedagogic theory might encourage others to

experiment further with their closer surroundings and use Living Labs as an educational asset. The environmental and economic (potential lower $CO_2$ footprint, and monetary cost, than other fieldwork activities) benefits of using campus-based Living Labs for educational activities may also be attractive for some and could play an important role in rationalising some programmes in terms of their economic efficiency and minimalising environmental footprints.

The case study presented here is from Keele University, a campus-based institution situated within the Midlands of the UK. We outline several HE teaching, learning and assessment activities from a range of disciplines, which use the Universities' 2.5 square kilometre rural campus as a multifaceted Living Laboratory (Fig. 1). Many of the activities students currently undertake are part of research and/or industry projects or are natural parts of the campus deliberately utilised as a learning tool. Keele University campus is host to academic, residential and commercial holdings; it has forests, fields, lakes, roads





and sports facilities. In short, it is a perfect analog for a small town. The nature of the campus therefore provides opportunities for a wide variety of educational experiences. Wider research activities are also included here in our description of Living Labs for education, and indeed, they are one of the most accessible learning environments on campus. These educational-based activities rely upon elements from the campus Living Lab which would often be in existence if they were part of a learning environment or not; this includes subject-specific research, industrial partnerships and research, plus

parts of the campus estate or buildings with particular purposes.

## 2 Living Labs Background

The Living Lab is a research area and phenomenon which has developed over the last few decades, first appearing in the early 2000s (Markopoulos and Rauterberg, 2000). It is a concept that has introduced new ways of managing and approaching innovations. A Living Lab allows innovations to be experienced and studied in an environment where people, the

environment, services, ideas and actions are manifesting in a natural and organic manner. Activities occur in real time, with experiments and studies exposed to a multitude of variables that would be impossible to simulate in a traditional laboratory setting. Multiple stakeholders are involved, and the dynamics of the research environment allows for research to be influenced by users in order to create new ways of working and/or deploying the technologies, concepts or ideas they are testing. Bergvall-Kåreborn *et al*. (2009) summarise the Living Lab as "*...an environment in which people and technology are*

*gathered and in which the everyday context and user needs stimulate and challenge both research and development, since authorities and citizens take active part in the innovation process*."

A Living Lab is essentially a partnership built between stakeholders, often public-private relationships (Bergvall-Kåreborn *et al*., 2009), where companies, organisations, authorities, public-groups and the general public can work together to create an

environment in which new concepts, services, technologies or policy can be tested and developed. A precondition of a Living Lab is that it is situated in a real-world context (Bergvall-Kåreborn *et al*., 2009). From this shared real-world, details of the innovation under scrutiny can be assessed, but, unlike a "sterile" or controlled lab environment, the results often transcend "discipline" boundaries and can be spontaneous and unexpected. Innovations can be tested for business case validity at the same time as function efficiency (of a technology for example) or social impact that the innovation may have.

This system means that the general public, and real-word infrastructure, play an active role in developing the innovative process. Living Labs have been viewed as different things by different authors; this is unsurprising when each Living Lab is likely to be constructed from different perspectives with different stakeholders, innovations and intentions. This makes a Living Lab a hard to define concept, although there is an emerging consensus as discussed by Hossain et al. (2019). Living Labs have been categorised or used as a type of environment (Ballon *el al.,* 2005; Schaffers *et al.,* 2007), a type of

methodology (Eriksson *et al.,* 2006) and as a system for enabling research (Bergvall-Kåreborn and Ståhlbröst ,2009). Liedtke *et al*., (2012) propose several research areas for the development of sustainable technology innovations within a Living Lab.





Several studies have looked at harmonising and collecting the various methods and approaches (Mulder *et al.,* 2007) or at producing concept designs for Living Lab implementation (Bergvall-Kåreborn *et al*., 2009). CoreLab (2007) suggest five principles in relation to Living Lab methodologies; these are:


- Continuity
- Openness
- Realism
- Empowerment of users
- Spontaneity

Because of the holistic nature of Living Labs, sustainability issues and "Grand Challenges" have increasingly become the focus of University based Living Labs (König and Evans, 2013; Robinson *et al*., 2013; Trencher *et al*., 2013 and Evans *et al*., 2015). These sustainability approaches are designed to make use of the cross-disciplinary nature of institutes and often 110 work with university estates, procurement or external consultants to provide projects within the Living Lab setting (Evans *et al*., 2015). The focus of this paper is more on the specific design of assessments and activities within the Living Lab learning environment as part of student modules from a variety of subject areas. Whilst not an explicit requirement, many of the issues tackled will fit within a broad sustainability umbrella as manifest through the United Nations' Agenda 2030 (United Nations, 2015) and the 17 Sustainable Development Goals (SDGs) covering the social, economic and environmental pillars 115 of society.

## 2.1 Living Labs as a Learning Environment

The Living Lab environment has a proven track record of producing valuable user-centric and technological/product information (see the European Network of Living Labs (ENoLL, 2020) for some examples) and this research approach continues to attract large amounts of funding and interest from well known companies (e.g. Siemens Ltd. at Keele, the Smart 120 Energy Network Demonstrator (SEND), the ground-breaking HyDepoly Project (the first ever demonstration of hydrogen as a fuel source in homes), and the Engie renewables development (Isaac, 2019 and Fogwill *et al*., 2020). The user-centric, collaborative, authentic aspects of Living Labs share many similarities with the pedagogic concepts of active learning (Prince, 2004; Settles, 2011; Freeman *et al*., 2014) and authentic assessment (Wiggins, 1990; Hart, 1994; Darling-Hammond and Snyder, 2000; Guliker *et al*., 2014). Authentic assessment is a widely contested term, first coined in the 1980s (Wiggins, 125 1990). It can encompass a wide range of different activities that require students to demonstrate higher-order thinking and complex problem-solving skills through context-specific tasks (Koh, 2017). It encompasses a range of applied and vocational activities and aims to engage learners in different ways. Although authentic assessment is not a new term, it has received renewed attention in recent years because of the increasingly diverse nature of our student body (OfS, 2020). Our students now transition to University with very different prior learning experiences, and different learning styles and



preferences. However, the ways in which we assess students has not kept pace with the increasing diversity of our student
        body (Darling-Hammond and Snyder, 2000).

The benefits of students undertaking authentic assessment, such as that provided by an extensive campus environment, and
interacting with real world examples and data, is well documented and closely linked to enhanced student engagement and
employability (Bosco and Ferns, 2014 and Senior *et al.*, 2014). Cumming and Maxwell (1999) suggest four key elements to
        authentic assessment:

- performance and performance assessment;
- situated learning and situated assessment;
- complexity of expertise and problem-based assessment and;
- competence and competence-based assessments.

The Living Lab provides an environment where these elements can all be met whilst the social-, economic-, product- or
concept-based focus of the experiment/test isn't compromised by the participation or actions of the students. Indeed, the
students can provide an additional stakeholder group or co-operate with a present stakeholder group (by collecting data for
        example). An additional benefit which can be part of Living Lab based education is how activities can expose students to
        thinking, processes and skills that they may not 'normally' be exposed to within a discipline 'traditional' curriculum. The
        Living Lab not only allows interdisciplinary working (such as geoscience, ecology and social science tangibly integrated)
        but lends itself moreover to effectively transdisciplinary working, in that knowledge and understanding are produced in
contexts of application (for a discussion of the various understandings of disciplinary prefixes such as multi-, inter- and
        trans- see Osborne, 2015). The five principles relating to the Living Lab methodologies (as highlighted above) and how they
        relate and contribute to the student experience and the learning environment are summarised in Table 1.

| Living Lab Principle | Contribution to the student experience and the learning environment |
|---|---|
| *Continuity* | This principle means that Living Labs are an ongoing process, a feedback loop, where innovations are experimented and evaluated in an often cyclical system superimposed on a linear time frame. The HE system of intended learning outcomes for students over subsequent years, with assessments and content which, whilst flexible, allows for assignment and assessment in a Living Lab to be planed over long periods of time. These assignments and assessments can be built into studies of long-term change. For students the ability to work on similar topics and problems across the levels of their learning nurtures deeper |



| | |
|---|---|
| | understanding and potentially increases engagement through deeper knowledge of a subject and the ownership brought by an ongoing project. Performance can be tracked through a students progress and complexities of experiences can increase. |
| *Openness* | For authentic experiences, the data gathered or used and the experience had by students needs to be open; this links strongly to the next principle, realism. By having open access to stakeholders, the Living Lab environment lets students explore authentic problems, some of which may have no solution. Data collected previously can inform teachers and learners and they can be open with their interpretations and analysis. By participating in the collection and analysis of data within a Living Lab, the students and their teachers will become a stakeholder group; as such, their activities must be open and their findings available in order to inform the Living Lab process. Authentic assessment advocates assessment which is showcased, celebrated, public-facing, used in practice and inherited (by stakeholders or the next cohort) |
| *Realism* | This principle makes being situated in a Living Lab an authentic experience, assessments and projects can reflect real-life tasks. Data gathered can inform research projects, including, but not limited to, the Living Labs "original" experimental aims. Assessment can be co-designed with stakeholders and presented to external audiences. This allows learning and assessment to be competence based and authentic. |
| *Empowerment of users* | By undertaking authentic experiences, learners are empowered knowing that they have a realistic experience of an environment. Once they have undertaken the task or assignment, they will have tangible knowledge to apply to their subject of study. This may manifest itself through the use of equipment, software, techniques or data types which the student has experience with, rather than simply learning about a topic. It may also manifest itself through the 'products' of assessment which are celebrated, showcased and have a legacy. |





| | |
|---|---|
| ***Spontaneity*** | Things change, they go wrong, results end up completely different to how we envisaged them to, it rains, equipment breaks, such is life. Spontaneous factors add to the complexity of tasks that students may be asked to undertake. They add elements of realism and situational and experiential learning. The spontaneity of a Living Lab can result in activities, data, skills, ideas and theories, and applications which transcend traditional discipline boundaries. |

Table 1: The 5 principles of the Living Lab (CoreLab, 2007) and how they can relate to student experience and the Living
Lab as a learning environment.

Tackling complex authentic problems requires a specific approach to teaching, learning and assessment. Practical problem-solving skills, which include collaboration, team-based, active and experiential learning, are key to encouraging the deeper learning required in order to develop the skills and competencies necessary to solve the problem (Espey, 2018; Kek and
Huijser, 2011). A deep learning approach fosters the ability for students to build on previous knowledge, to draw on experience, to bring together disparate information and organise it into a coherent whole, to identify relationships, to form hypothesis and ultimately enhance conceptual understanding (Biggs, 1987; Ramsden, 1992). Aligning the assessment and teaching method through the construction of related learning objectives (Biggs, 1996) allows for the critical thinking skills to be embedded throughout the teaching. Studies by Brodie (2009) and Yuan et. al., (2008) conclude that a higher level of
critical thinking skills are found in students who have experienced a problem-based learning environment.

Identified as a teaching method which provides a good example of constructive alignment (Biggs, 1999), problem-based learning (PBL) is a socially constructed pedagogy whereby all students are involved in the co-construction of knowledge based around self-directed learning, groupwork and the exploration of problems (Kek and Huijser, 2011). Originating in the
1960's from McMaster University in Ontario, Canada, it was first used to teach students within the medical profession and its many constructs are well described by Barrows, (1986). At Keele University, a hybrid-PBL model was developed to ensure that delivery would be feasible with regard to both time and staff constraints (Bessant et. al., 2013). The hybrid-PBL form follows a blended learning approach containing a mixture of PBL face-to-face groupwork, online screencasts, traditional lectures and visiting professional case study speakers. This allows critical thinking skills and discipline-specific
knowledge to be developed simultaneously (Espey, 2018; Kek and Huijser, 2011). Within this mode of teaching, the lecturer assumes the role of facilitator, guiding but not prescribing the learning.

Examples of Campus as a Classroom activities conducted at Keele University which are building on existing components or activities are given below; these activities were chosen to highlight the breadth of opportunity for Living Labs for education.
Having students learning, and staff teaching in these environments adds an additional stakeholder group to the Living Lab.





The benefits of this group include: insightful feedback to processes and products; a level of expectation for procedures and experiences to be authentic; fresh perspectives and outlooks on projects each year; and, the potential for certain stakeholders to influence, educate or expose potential future consumers or employees. The campus also acts as a test bed for new methodologies to be approved or for academic research to be conducted - these activities benefit from having students and
staff as a stakeholder group in the same way that any partner company might do. Each of the given case studies outlines the activity undertaken, some of the logistics involved or processes used, and how the activity sits within a Living Lab e.g., how activities add to user-centric, industrial, research and other activities.

**2.1.1 Campus as a Classroom Case Study 1: Environmental Baseline Survey**

An Environmental Baseline Survey (EBS) Module was created in 2006 and aimed to increase the employability, field and
research skills of FHEQ Level 5 (UK Government, 2020) Physical Geography students at Keele University. The module was developed in collaboration with MJCA Environmental Consultancy to ensure that students completing the module gained key skills required for graduate jobs in the environmental and geoscience sector (Robinson and Digges la Touche, 2007). The module uses the Keele University Campus as a Living Lab to undertake student-led, experiential, active learning and to provide students with an authentic experience of collecting, analysing and presenting data. The module also aims to teach
students about a range of techniques relevant to research and data collection in their degree programme, to be more critical of authentic research scenarios and build upon existing skills developed during FHEQ Level 4.

Working in groups, the students are set the brief that that they are acting as Environmental Consultants and must undertake an Environmental Baseline Survey on the soils, habitats and hydrology surrounding the Keele lakes (Figs. 1 and 2) on Keele
campus. Every week for 8 weeks the students spend 3 hours collecting and analysing specific data to write up as an industry standard EBS. Each class begins with a short briefing session, outlining the aims and objectives of the practical and some background information (e.g., risk assessments, maps and methodologies). Following this, students go out into the Living Lab to watch a demonstration of the techniques and equipment to be used by staff. The staff also spend some time asking questions to the students to get them to think critically about how best to sample (limitations and number of samples to be
representative). By revisiting and building on existing knowledge in the introductory sessions, promoting discussion and reflection in the field and having the emphasis on active learning, the sessions foster deep, reflective learning in an authentic environment (Bloom 1956, Russell *et al.* 1984; Ryan and Deci 2000; Light and Cox 2001).

The students gain experience of a range of techniques including water chemistry sampling and analysis (major ions, pH, EC,
temperature) of the lakes and inflows, groundwater measurements (using a network of piezometers installed around the lakes), discharge readings using dilution gauging, soil sediment analysis (description and logging, loss-on-ignition and water content, grain size analysis), and surveying techniques to produce geomorphological maps. The students also make use of the Keele Meteorological station (used by the Met Office, UK), which collects data every hour, to make interpretations of the




water chemistry, discharge, groundwater and soil data. Students can also compare their results to data collated over the last 4
years by past students to analyse whether trends are comparable. Using the extensive woodland and grassland environment,
skills in habitat survey methods are also included. This follows the JNCC (2010) Phase 1 Habitat classification system,
currently a key component of Environmental Impact Assessments within the planning system in the UK (Joint Nature
Conservancy Council, 2010). The Living Lab provides an opportunity to collect ecological data within the framework of an
environmental baseline assessment, thereby providing an authentic understanding of the role that ecological data can play in
protecting biodiverse sites within the planning system.

The EBS is assessed through a group-led, industry standard Environmental Baseline Report which must collate, present and
analyse all groups' data to produce a professional report, as a graduate would be required if working specifically as an
environmental consultant or geoscientist but more broadly in any analytical career. The module therefore caters not only to
students who wish to go into the environmental/geoscience sector but has many transferable and desirable skills that
graduates can take forward into their careers and FHEQ level 6 studies.

The module is supported by self-guided, asynchronous online resources (see *The role of digital/virtual platforms in Campus
as a Classroom/Living Labs* below) related to the techniques covered each week. These resources are intended to be
completed independently and include short videos, core texts and examples of academic research using the techniques
covered in the practical. Additionally, there are self-directed worksheets encouraging students to learn and focus on key
definitions and concepts. These resources are designed to promote knowledge but to develop critical and reflective thinking.

### 2.1.2 Campus as a Classroom Case Study 2: Simulated Crime Scene Investigations

Created on campus in 2008, as part of a funded Teaching Innovation Project, in collaboration between academics and the
Keele University Estates team, a simulated multiple buried victim crime scene was created within a secure area, with ethical
approval given by the University and by the Department for Environment, Food and Rural Affairs, UK (DEFRA).

For geophysics-based FHEQ Level 5 or Level 6 (UK Government, 2020) undergraduate modules, the outdoor practicals
involve a student group-led, problem-based scenario, involving them being a ground forensic search team, tasked with non-
intrusively investigating a specified search area to locate (and characterise if possible) buried murder victim(s) (Fig. 3), for
then hypothetical intrusive investigation teams to confirm the presence/absence of victims at locations specified by the
students. This style of problem-based, active, outdoor practical learning has been proved to really accelerate learning and
understanding and greatly enhance students' employability skills (see Murphy and Pringle, 2007; Pringle *et al*. 2010). The
forensic search angle has also proven useful to enthuse and keep students engaged on the task in hand.





Students are provided the opportunity to design a robust forensic search strategy by choosing their own search methods/equipment to use what they have learnt theoretically in class. Each group are then collect the multi-disciplinary site data in a time-limited period on the campus site, before subsequently processing and integrating datasets back in the lab, to produce a technical group report with recommendations on which area(s) to intrusively investigate, as would be the case

when doing this for real. Supervisors have direct experience of this and are on hand to discuss and solve any problems as they come up, but it is emphasised that this is a student exercise and so they are free to make (and hopefully correct) their own mistakes. Many of these graduates go on to related commercial careers using the skills learnt here, especially within the geotechnical site investigation industry.

Module *intended learning outcomes* include: (1) to work effectively as part of a student-led team to solve a geoscientific problem within a limited time frame, use critical thinking, multi-disciplinary data analysis and interpretation and, (2) to use technical writing, numeracy and computing skills in the context of forensic geoscience investigations. Student marks for this formative assessment usually average above their other module components, with end-module evaluation quotes which are almost universally positive, including "*Practicals allowed independent thought & organisation*" and "*Practical session*

*interesting & fun to carry out*".

The site has also been used for over 10 years as a collaborative research environment between students and staff. This has been both as formative assessment, as part of their under- or post-graduate courses, as research projects, or indeed as non-credit bearing collaborative research projects. Student project-led examples that have been published in international-

journals include: (1) determining if magnetic surveys could detect buried victims (Juerges *et al.* 2010), (2) looking at seasonal factors affecting forensic geophysics surveys (Jervis and Pringle, 2014), analysing soil water from such graves (Dick and Pringle, 2018) and the long-term geophysical monitoring of the site (Pringle *et al.* 2016 and 2020). Research outcomes have directly led into refining UK and international Police search strategies, allowing them to compare live missing person and unsolved cold case data to controlled data (e.g. Pringle and Jervis, 2010) and even test their search

strategies.

Finally, it has also been used for many years as part of our school's very successful outreach and engagement strategy, from having local schools visit to conduct a simulated forensic investigation, to having Nuffield Foundation Placement School and FE college students since 2010 using it as their 4-week summer placement collaborative research project.


During the recent COVID lockdown, outdoor laboratory practicals could still be run in certain situations, and thus socially-distanced students still attended, used gloves on equipment and facemasks when collecting data, with subsequent data processing occurring remotely but collaboratively through Microsoft Teams. When practicals could not be physically run at all for students, a virtual practical of this case study was generated within the Thinglink online platform, with short, digitally-





recorded videos illustrating how each dataset was collected by different equipment, before the datasets were provided and again virtually processed remotely.

### 2.1.3 Campus as a Classroom Case Study 3: Greening Business: Employability and Sustainability

Since 2008, Keele University has run the 'Greening Business: employability and sustainability' as a flipped-classroom module for Level 4 students from any degree pathway at the University (Robinson, 2009). With a strong emphasis on

fostering the skills required to drive forward positive environmental change within their future workplace, the module has a core transformative agenda which allows students to consider their own perspectives, attitudes and values in the context of their relationship with the business world. As the sustainability agenda continues to gather pace internationally, especially with regards to climate change and net zero carbon targets, the role that businesses and large organisations play in helping to achieve the global Sustainable Development Goals (United Nations, 2015) becomes ever more important to address.

Equipping professionals of the future with the skills and understanding to engage successfully with complex, multidisciplinary, real-world sustainability problems is a key aim of this module and this provides a genuine and fundamental link to the Living Lab learning environment at Keele University.

All organisations and businesses are required to address their environmental and sustainability impacts, thereby making the

learning relevant to all students whether they are environmentally-conscious or not. Behind the scenes of every large business, whether forced by international law or regulated at a national level, lies a complex web of voluntary and regulatory compliance, monitoring and reporting, all addressing the ways in which their work affects the natural environment (e.g. carbon reporting / waste transfer / ecological impacts of development). The social agenda is equally important, with companies being mindful of ethical issues within their supply chain from slave labour to fair wages or the right of workers to

form a union. All of us act as stakeholders within the business world, whether we are aware of it or not, and this 'stakeholder lens' becomes a powerful tool for encouraging students to explore the barriers and opportunities for improved sustainability performance within this sector.

The hybrid-PBL model used on this module gives students the chance to investigate one of these issues in more detail,

whether it be an operational issue, or one focused on behaviour change with businesses, departments and service providers located within the Living Lab environment on campus. Many of these projects are complex and open-ended with no single solution, and the learning is less scaffolded than in traditional PBL models (Barrows, 1986).

Students present their completed project to a panel of relevant stakeholders in video format, followed by a Question-and-

Answer session. Past findings have been used to develop projects on site, illustrating that this truly is an authentic form of assessment and that the University is genuinely interested in their findings. The projects are set within a loose structure within which students are responsible for organising group roles and drawing up an action plan; identifying and interviewing



professionals who can assist them with their enquiries (eg. environmental manager, estates or catering staff); gathering baseline data about the issue they are investigating (e.g. how much waste is produced on campus; how many students and
staff commute in single-occupancy cars); and linking their issue to existing over-arching corporate strategies, relatable targets or visionary statements. They also need to develop a storyboard, film and edit relevant footage; develop the narrative, and present their findings to the assessment panel.

During the Covid pandemic, in-situ groupwork changed to a blend of synchronous online sessions, supported by live lectures
and asynchronous video materials.

Their presentation must include links to the SDG's and the inclusion of clear recommendations linked to SMART targets. Previous projects have included the development of a communications strategy for the Keele Student Union new Zero Waste shop or exploring packaging issues with an on-site book retailer; working with estates to propose land management practices
which encourage pollinators or hedgehogs; working with the Head Chef to look at students' relationship with food choices and related waste on site, and developing ideas for plastic take-back schemes. More detail regarding project options is provided in Table 2.

---

**Living Lab Project options as part of the Greening Business Module**

**Transportation**: investigating the most 'sustainable' options for commuting staff and students including bike hire schemes, reduction in single-car use and sustainable travel strategies;

**Recycling practices:** investigating opportunities to enhance use of the different forms of recycling on campus including issues with effective messaging to ensure the correct separation of waste, developing plans and activities to reduce single-use plastics;

**Communication and messaging strategies:** investigating how students could better engage with the sustainability initiatives on campus including the Zero Waste shop, Switch-Off energy initiatives, the Great Donate initiative; the Smart Energy Network Demonstrator (SEND), the ground-breaking HyDepoly Project and engaging students with the Climate Emergency (declared at Keele in 2019);

**Carbon reduction;** investigating the opportunity for carbon sequestration focusing on the natural environment found on campus, exploring opportunities to reduce the carbon footprint of field courses;

Food and growing initiatives; investigating the links between consumer choice, food options and climate change; exploring opportunities to engage more students in growing food on campus;

---





> **Biodiversity;** investigating the use of Biodiversity Off-setting and rewilding schemes related to on-site development; developing management plans for specific wildlife species on site;
>
> Diversity and equality: investigating ways in which opportunities can be improved across the campus;
>
> **Health and wellbeing;** investigating the potential to further develop outdoor fitness activities on campus
>
> **Education for Sustainable Development:** investigating opportunities to embed the sustainable Development Goals (SGD's) into the curriculum
>
> Specific sector improvements; investigating how the on-site bookshop or the events and conferencing team could better embed sustainability in their operations;
>
> Auditing of practices on site; investigating energy use or waste generation on campus.

Table 2: Living Lab Projects – options developed as part of the Greening Business Module


The purpose of running educational projects within a Living Lab in this way, is to create the atmosphere for collaborative learning whereby learners co-construct their own knowledge, ultimately generating new sustainability knowledge. Other skills are also evident; team-working; critical thinking; negotiation; listening; communication; presentation skills; awareness of ethical and value-based motives; a wider understanding of global citizenship and reflection. Being an elective, this module

provides the ideal opportunity for students to work with an intercultural team, in an interdisciplinary environment, which encourages discussion from different perspectives and places of understanding, modelling the real-world environment that they may one day find themselves working in (described as the principle of Empowerment of Users by CoreLab, 2007, see Table 1). Interdisciplinary and intercultural modules and projects such as this example might provide a useful start for discussions to Decolonising the Curriculum for programmes engaging with the initiative.


### 2.1.4 Campus as a Classroom Case Study 4: Drone Technology

The use of drones in Earth Sciences and cognate disciplines has grown exponentially in recent years, both within academia and in industry (Luppicini and So, 2016), so it is an employable new sector that currently has a lack of trained professionals (King, 2014). Therefore, providing graduates with an authentic opportunity to obtain the necessary skills to pursue a career

in drone technology is something that can enhance their employment prospects. Thanks to the experience of some of the Keele staff using drones as part of their research activities (Nobajas *et al.,* 2017) and to funding obtained from a variety of sources, a series of new teaching activities were designed in order to allow students to have a realistic, spontaneous experience using drones.





The biggest limitation when introducing drones to undergraduate students is safety, as a lost drone can cause both material and personal damage (Stewart, 2016). On top of that risk, there are a series of legal limitations that need to be taken into account; not complying with these can result in hefty fines and even jail sentences (Stoica, 2019). For example, such regulations mean that drones cannot be flown near houses, roads or groups of people. Therefore, due to safety and legal concerns, finding an adequate area to carry out the practical sessions is of paramount importance, and Keele's University

campus offered the ideal location to take the classroom outside and practice the flying skill learnt inside. In this sense, running a similar type of activity in a city-based campus might be difficult, as regulations and health and safety concerns would make finding a location a challenge, although the use of park or recreation ground could be negotiated.

In Keele's case there was a choice of several locations to choose from. Over the years the practical flying sessions have been

carried out in a variety of environments, but it has been found that the best location is one of the most remote areas of campus, which is currently not developed and offers easy access, essentially no traffic of any kind and considerable distance from any buildings, so it complies with drone flying regulations. There is always the risk that the area may be developed as the university grows but, since there are other places within campus such as sports fields or other open areas that are also suitable for the activity, this should not pose an insurmountable problem.


As part of an FHEQ Level 6 GIS module, students are initially introduced to the school's fleet of drones and the different characteristics and elements of each drone are presented to the students. Once they are familiar with all the controls and technical details a programmed flight is carried out. Programmed flights are key to working with drones in a professional way, as they allow performing automated actions that lead to a photogrammetric output, that is to say, an aerial image that is

geometrically correct (e.g. Nobajas *et al.,* 2017 and Priddy *et al*, 2019). In combination with theoretical and computer-based sessions, the gathered data are processed using Structure from Motion (SfM) software and an accurate 3D model is generated (Nobajas *et al.,* 2017). All these steps help the students understand what can be achieved with modern drone technology, a process that is widely used in a variety of industries such as crop production, surveying, mining or archaeology (Reinecke and Prinsloo, 2017). For example, Keele staff discovered a medieval Templar-built road on the outskirts of campus thanks to

the use of drone technology, and this is used as part of the teaching materials (Burnett, 2018).

Finally, students are given a small drone each so they can gain hands-on experience on how to fly a drone. In order to minimise financial losses and reduce damage to property or the public, very simple, cheap and light (~ 100 g) drones are provided, as they are so nimble any crash has minimal consequences. Students are encouraged to practice as part of the

practical outdoor session until they get comfortable flying their drones. Once the teaching session is over, they are allowed to keep the drone for around a month and they are expected to take an aerial photograph (Fig. 4) with it that complies with all the legal limitations imposed on drones. Although they can take the drone wherever they want to, most of the photographs





submitted are taken on campus, as, apart from being quite picturesque, it has all the necessary characteristics that make flying a drone a safe activity.


None of the described activities are directly assessed. The contents taught during the lectures, computed-based practicals and the outdoors sessions are part of the materials assessed as part of a test in the module. The drone photography students are expected to obtain is entered into a photography contest with a prize given to the winner. Students have had a very positive attitude towards these activities, which have won two teaching-led awards.

**2.1.5 Campus as a Classroom Case Study 5: COVID19 Fieldwork**

The emergence of the Covid-19 pandemic during the 2020/2021 academic year forced a rethink concerning the possible locations from which undergraduate fieldtrips could be safely delivered. The campus provided multiple opportunities to investigate geographical and environmental topics right on our doorstep, reducing the need for travel and preventing the need for residential fieldtrips which could not be undertaken at that time. Home to two different lake systems, small rivers,
different blocks of woodland and acres of grassland, the campus itself became the fieldtrip host.

Climate change is a core teaching and learning theme within the school, and the University campus provided the opportunity to explore past environmental change linked to previous glacial events (Fig. 5), as well as contemporary issues such as forms of renewable energy generation. Situated on the geographical edge of the last glacial maximum of the British Irish Ice Sheet
(Clark *et al.* 2017), the campus provided students with the opportunity to explore glacial geomorphology and sedimentology firsthand using a mixture of GIS mapping techniques and inland field sections. Clast shape and roundness counts were used to investigate the transport pathways of the stones found in the field (Evans and Benn, 2021). The presence of a new on-campus renewable energy installation had revealed numerous glacial erratic clasts within the superficial sediments, and these were used as a known dataset with which to compare those found across the rest of the campus. Students had visited the
renewable energy construction site previously when trenches were exposed for archaeological investigations, but this fieldtrip provided an opportunity to revisit the site during the installation of the solar panels themselves – a fitting link to the modern element of the climate change theme. The campus was also used to explore wider implicit and explicit sustainability messaging (Djordjevic and Cotton, 2011), using self-guided materials to identify and classify different messages seen on site, including the role that the 'hidden curriculum' plays on campus (Orr, 1993).

**3 Discussion: Campus as a Classroom Concept Design**

The concept design for Campus as a Classroom activities ultimately relies on two factors: 1) activities, projects and infrastructure available to the designer, and 2) an intended learning outcome of the proposed task. Other elements to consider include logistics, costs (if any), ethical implications of using the Living Lab as a classroom, and safety issues. The broad design for a Campus as a Classroom activity needs to cover the following things:





interactions, experiences, processes and a guiding strategy. These activities can be planned much like other assessments (see
Wiggins, 1998), where student well-being, preparedness and learning outcomes are understood and acted upon. On a broader
level, the design and implementation of these activities is informed by the Keele Social Curriculum and Curriculum Design
Framework (CDF; Keele University, 2021). This framework sets out key principles for innovative programme design, within
the broad themes of Digital Education, Sustainability, and Health and Wellbeing. The Campus as a Classroom concept offers
some valuable synergies with the themes and subthemes of the framework. For example, the learning activities delivered in
case studies 1 and 2 in this paper both adhere to the subthemes of Authentic Assessment and Employability and Civic
Engagement, assessing learners (formatively or summative) through the application or real-world, practical skills that are
critical to the respective career pathways associated with each. Furthermore, both case studies draw on the subtheme of
Technology Enhanced Learning, making use of asynchronous digital resources and media to support learning. Their student-
led approach also ties in with the subtheme of Inclusive Learning, allowing more flexibility for learners to engage with the
process in their own way. Case study 2 also offers direct civic engagement, with a significant contribution to outreach and
local engagement projects. Case study 3 synergises primarily with the subthemes of Employability and Civic Engagement,
and Global Perspectives, offering learners not only an opportunity to engage with real-world challenges, as well as the
chance to explore diverse backgrounds and experiences within the realms of business and sustainability. Finally case study 4
exemplifies Technology Enhanced Learning and Employability and Civic Engagement, allowing learners to access high-end
drone technology (and associated computer modelling programs) and develop skills in a sector within which expertise is in
high demand.

Using a Living Lab does potentially increase the time and effort required in the planning and set up of HE teaching and
learning activities and assessment. New learning environments and their inherent safety issues must be considered and the
students participation in the lab must be carefully considered including such questions as; are the students suitable
stakeholders? Does their participation as stakeholders change any processes or ethical considerations of the Living Lab
experiment? A concept design framework - based on four strands - is suggested. This concept design provides a structured
foundation which ensures high-quality assessment and/or activity planning within a Living Lab environment. Such
frameworks provide useful guides for development whilst highlighting technical, logistical and practical considerations of
what might be achievable and appropriate.

The Campus as a Classroom concept design framework is divided into the following five strands. The first four are provided
to ensure activities within a Living Lab environment are fully considered, practical and beneficial to students. The fifth and
final strand outlines the importance of embedding Living Lab learning activities into the wider curriculum and providing
students with proper preparation and support in the learning activities they will be undertaking.



### 3.1 Campus as a Classroom Framework

#### 1) Guiding Strategy

What is the purpose of the Living Lab which is to be used as the basis for this activity? Does the activity compromise this?
Most importantly, are the pedagogical benefits of learning in a Living Lab setting being considered? Pedagogical innovations must be included as an educational innovation, and not as an innovative tool for the sake of using that tool. Assessments in Living Labs should provide authentic experiences which allow for spontaneity and openness in purpose and resulting information gathered. Before embarking on an assessment/activity within a Living Lab environment consider the synergy of said activity and the purpose of the lab itself; neither should compromise the efficiency and aims of the other.
Institutional educational vision and strategy (e.g. the CDF, Keele University, in our example) should also be considered at this point.

#### 2) Interactions

What are the teachers and students going to do - who or what will they be interacting with, is this a passive process or an
active process? Do you need permission for the work to be undertaken or ethical considerations to be made? Health and safety of the activities must be considered, for the participants but also other stakeholders and the environments they are working in. The type of interaction should also be considered. For example, students may be interacting directly with other stakeholder groups or the labs innovation (i.e. a piece of technology being tested), or students may be acting passively within the Lab. The level of interaction therefore dictates the influence the activity may have within the Living Lab, this has
impacts on aspects of the activity such as feasibility, logistics, safety, impact and overheads etc.

#### 3) Experiences

What activities will the students undertake, what are the links between the activities and the intended learning outcome? Are there logistical considerations with equipment or with getting to the intended area of work? The skills, competence and
aptitude of the student cohort need aligning with the activity, prerequisites or prior learning should be mapped to the proposed activity.

#### 4) Processes

How will students gather data, who owns this data and what will be done with the data once the students have used it? Are
the students going to feedback into the Living Lab exercise or passively interact, gaining skills and experiences but not becoming active stakeholders?

#### 5) Embedding Campus as a Classroom/Living Labs within the curriculum and providing support for learning activities



Very few educational activities work in isolation. Most are best suited to a blended approach where a mixed modality of teaching delivery is provided. This might include asynchronous or synchronous delivery of materials in a variety of environments (both in situ and digitally). Nearly all authentic activities will require some form of preparation, including training with equipment, contextualisation of the activity, introduction of key concepts and theory, and provision of fundamental health and safety information, all of which provides a foundation upon which learners can build. An efficient way of providing some of this is via asynchronous digital resources, which allow students to use them before the activity, but also during the activity should they need to. This has the potential not only to enhance the efficiency of learning, but also to enhance greatly the accessibility and inclusivity of learning, particularly where learning is dependent upon access to specific facilities or resources such as analytical instrumentation. It is therefore a critical part of developing these resources to plan and develop any additional teaching materials required as the Campus as a Classroom activities are developed. Examples of materials which we have found to work well include: synchronous discussive sessions (recorded for flexibility), lab based sessions, pre-recorded video material, virtual reality introductions to the work environment (Rogers 2020), and digital/virtual lab equipment.

For example, here at Keele University we have experimented with the supplementation of traditional laboratory-based teaching (microscopy, XRF spectroscopy, and ion chromatography) with virtualised forms of the instrumentation in question. These learning resources together constitute a variety of virtual laboratory (e.g. Mercer et al., 1990; Koretsky et al., 2008) which aim to provide background information on specific techniques, and to simulate the running of the instruments themselves (e.g. calibration, data collection). Simulated laboratories have found application not only in the traditional perception of a chemistry lab, but also in physics, chemistry, computer science, biological science, material science, and engineering (see Jeffery, 2021 and references therein). Their application has increased in recent years due to the ever-increasing technological developments available to educators. The actual nature of a virtualised laboratory can range considerably in scale and scope, from compact and simple materials designed to meet very specific learning outcomes such as those found in an individual class (e.g. Jeffery et al., 2021) to materials designed in full 3-D environments and/or covering a considerable range of academic material (e.g. Hernández-de-Menéndez et al., 2019 and references therein). Although there is no real consensus on the validity and value of virtual laboratories, they are regarded to have the potential to enhance or support the following key factors: the development of the learner's key skills and academic performance (e.g. inquiry skills, practical skills, perception, communication skills etc., the learner's motivation and mental wellbeing (e.g. provision of virtualised laboratories as a supplement to learning can reduce or mitigate anxiety in distance learners), and the efficient use of education resources, including financial implications for the educational institute and the time required for the educator (e.g. face-to-face, hands-on teaching time could be reduced using a virtual lab as a preparatory learning activity; see Jeffery 2021 for a review). Nevertheless, their application may lead to negative effects, such as the potential discouragement of learning using real instrumentation, or reduced interaction between learners and teachers or other learners, increased risk of



plagiarism, and reduced opportunity for the development of physical skills (Chan and Fok, 2009). The investment of time required to create such learning materials may also impinge on their abundance or quality (e.g. Wästberg et al., 2019).

Education in these areas has previously been subject to a number of potential barriers to learning. For example, the amount of time that an individual learner can spend developing hands-on experience with a given instrument is dependent on the availability of access to the instrument (e.g. instrument to learner ratio), as well as the face-to-face time required with an appropriate teacher. For many higher-end analytical instruments, there may be only a single instrument available and so learner access, and therefore their ability to develop practical experience, may be heavily restricted to specific and limited

times. There may also be health and safety considerations which prevent learners from using an instrument, which may be derived from the instrument itself (e.g. X-Ray fluorescence spectroscopy), or may be linked to external factors (e.g. COVID-19 epidemic). Finally, it must be considered that there may also be individual disability-related needs which make it difficult for learners to access instrumentation. Under the Equality Act 2010 (Legislation.gov.uk, 2010), educators are obliged to provide reasonable adjustments and ensure that materials are accessible for learners with additional needs. To this end, we

have found that the application of asynchronous virtualised forms of the instruments given above has been viewed as favourable to learning by learner and educator alike, by providing learners with a means of exploring an instrument or technique at a time and place of their choosing, in an interactive and accessible form (Jeffery et al., 2021). These resources therefore have the potential to provide a powerful complement to Living Labs, adding depth, diversity, and flexibility, particularly when treated in a supplementary fashion (e.g. Sancho et al., 2006; Bean et al., 2011) rather than being mutually

exclusive with traditional lab-based teaching. Nevertheless, their creation and implementation should be considered carefully to maximise compatibility with existing learning narratives.

Finally, a Living lab on campus may hopefully go some small way to address some of the inclusivity and diversity issues within applied environmental, geography and geoscience courses that have been highlighted recently (e.g. Dowey et al.,

2021). Offering authentic and meaningful alternatives which can reduce prohibitive residential course costs (where HE institutions charge) and the requirements for very robust and expensive student field gear and equipment. If students (and/or staff) don't enjoy or are unable to attend long periods away from their university or home base (have family or care obligations, etc.) Campus as a Classroom ensures those students do not get inferior 'paper based' (often literature review or essay style) alternatives and can participate in authentic activities.


**Conclusion**

Living Labs and campus-based activities – Campus as a Classroom - can be used to provide authentic learning and teaching experiences for HE students. The outdoor environment is perfect for getting students to use field equipment and allowing



them to work in an environment where spontaneity and the opportunity for things to go wrong, or not as planned, as well as
then solving these issues is showcased here as being a very important learning experience. Campus as a Classroom gives
students interdisciplinary experiences and allows the application of information disseminated by other teaching and learning
methods within the curriculum. Activities within a Living Lab should be framed by: 1) a Guiding Strategy of *why* a Living
Lab is appropriate; 2) clearly outlined interactions (with people or things) including health and safety consideration; 3) a
clear idea of the Experiences, Intended Learning Outcomes and activities to be undertaken; 4) an idea of Processes such as
how student derived data will be collected and if it will be used in the Living Lab; and 5) a clear scaffold of supporting
material preparing students for activities within the Living Lab. Making use of the Campus can also help ensure course
Intended Learning Outcomes are met, whilst potentially reducing carbon footprints (by not travelling to external locations,
for example) which is more sustainable and thus to be encouraged. These types of activities may be a more inclusive option
for students either not wishing, or unable, to go on residential field courses. Campus as a Classroom activities can also result
in student-led innovations being implemented across the place they study, heightening student empowerment and including
students as stakeholders of the environment in which they learn.

**Author Contribution**

The drawing together of the co-authors experience and teaching activities into a framework and corresponding manuscript
was conceived by SLR. All authors have contributed to case studies herewith and have all contributed to the drafting of the
manuscript.

The authors declare that they have no conflict of interest

**Acknowledgements**

Our colleagues from the School of Geography, Geology and the Environment at Keele University are thanked for sharing
their experience and expertise in using, developing, designing and running campus-based learning activities. The Keele
Estates team are thanked for logistical and site maintenance support for case study 2, and wider campus activities.

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





**Figure 1: A basic map of Keele University campus in Staffordshire, UK, highlighting some of the areas forming parts of the Living Lab (see key and text for details). EBS = Environmental Baseline Survey (satellite image from © Getmapping Plc (Digimap)).**






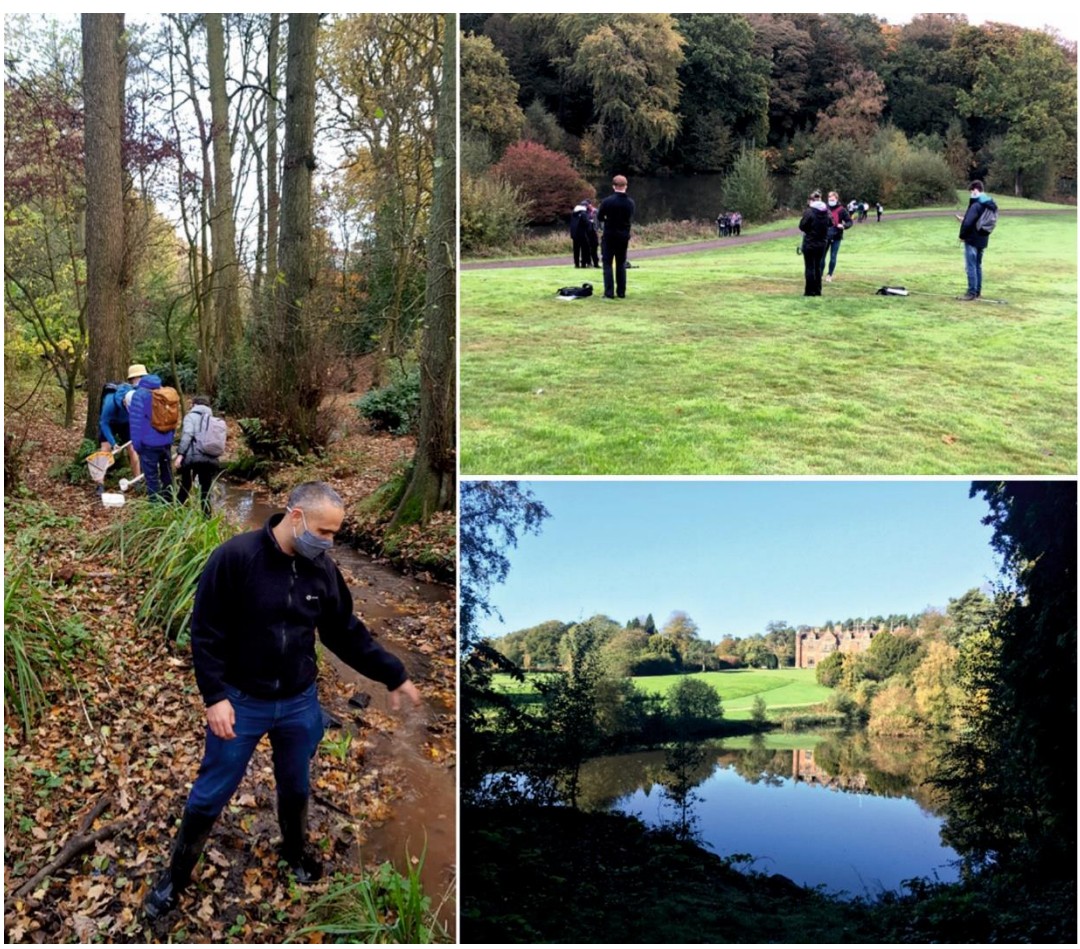

**Figure 2: Images of (left) river sampling, (top) soil sampling and (right) Keele lake sampling elements of Keele University campus used as part of the Living Lab where students collect samples and data as part of an Environmental Baseline Survey.**





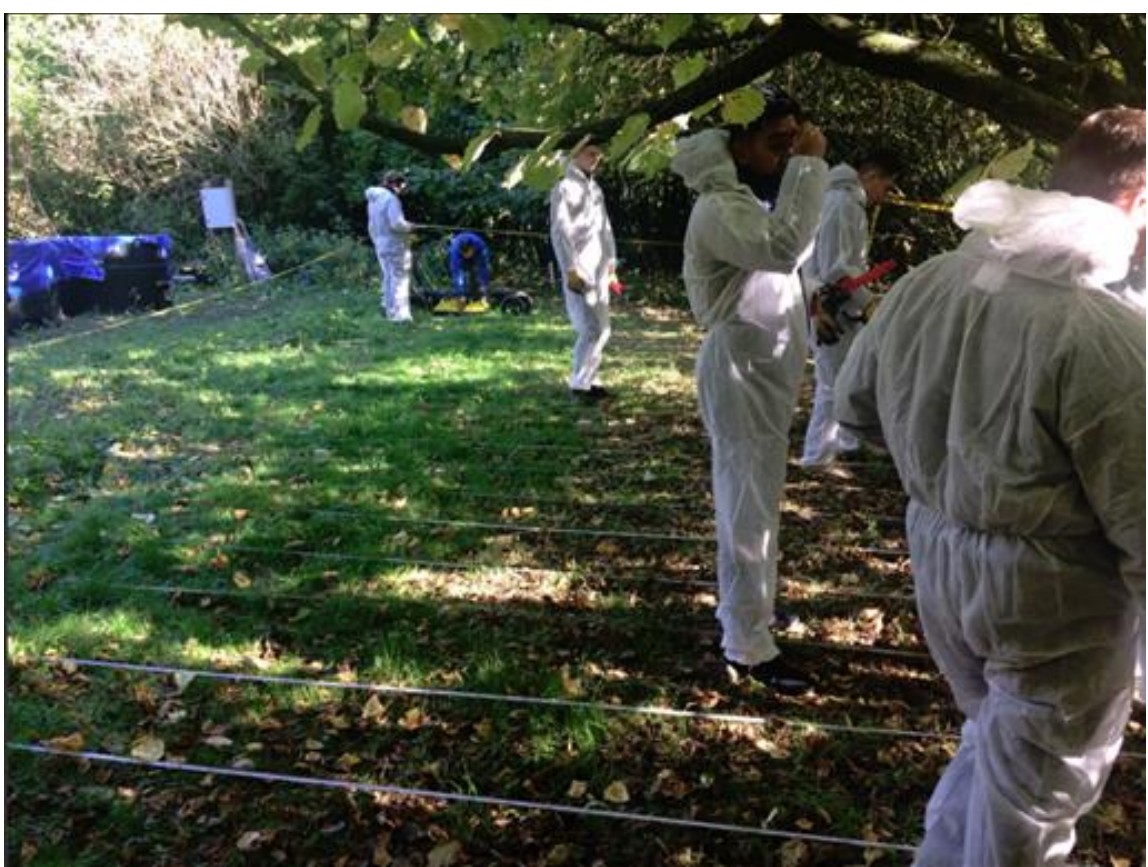

**Figure 3: Keele University undergraduate students collecting near-surface geophysical data over a simulated crime scene site on**
**campus. Students conduct the activity using the same methodologies and equipment (including clothing, as seen in the image) that**
**a near-surface geoforensics search team would in active casework.**





**Figure 4: Drone-collected images taken on Keele University campus near Keele Hall (see Fig.1 for location) by a student in 2017 using one of the School-provided drones. Most students decide to practice their drone photography skills within the campus limits**
**as it offers a safe and legal environment to hone their piloting abilities.**




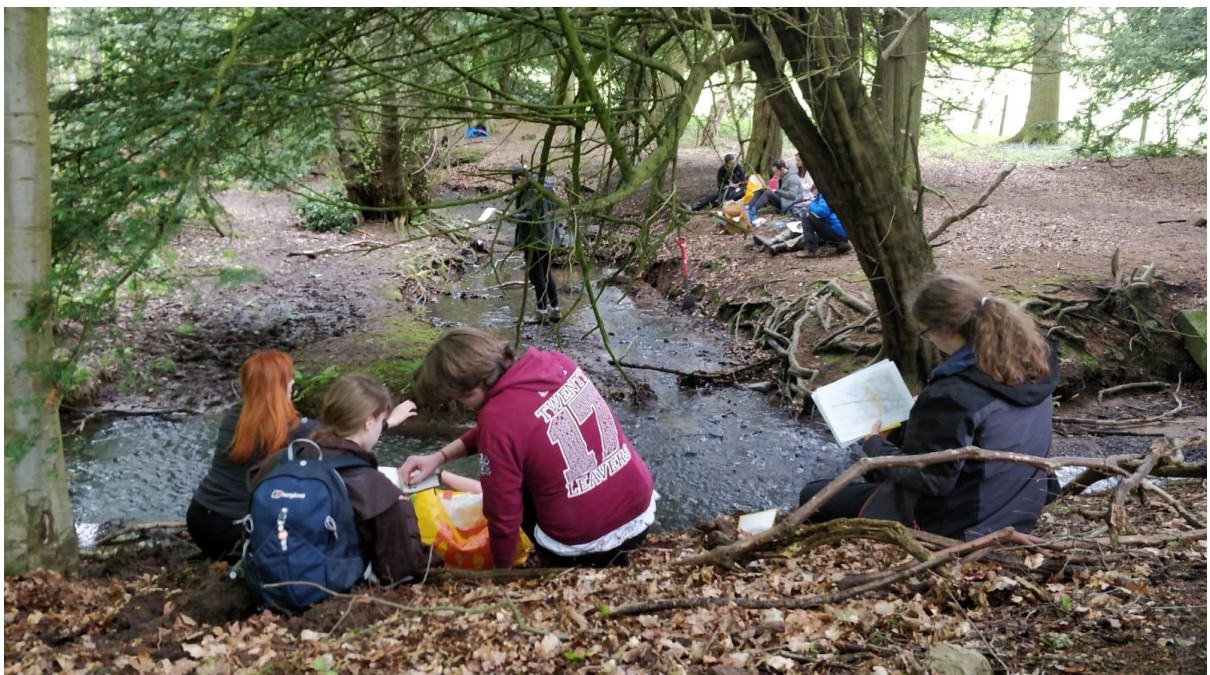

**Figure 5: Keele University students investigating glacial sediments as part of an on-campus fieldtrip during the Covid-19 pandemic.**