# Peer review of "The University Campus as a Learning Environment: the role of a Campus-based Living Lab in a Blended Teaching and Learning Environment"

_Geoscience Communication, 2021_

## Author Comment (AC1)

Dear referee,

Many thanks for the time you have taken reading and commenting on this submission. In order to make our response more focused we have provided comment (in red) directly to each point:

Dear Authors,

Thanks for the opportunity to read and review your work on living-labs. The papers case studies and framework for using a living-lab will be of interest to educators and practitioners in the field. Although I am supportive of the paper and can see its potential, I do have some comments and suggestions that I outline below. These are intended to make the paper stronger, not detract you from sharing your work in this area.

Many thanks, we agree that there are areas that can be strengthened and pleased you can see the submissions potential.

1. Living Labs - There is a short definition of living labs in the introduction and a broader discussion later in the introduction. I feel this needs to be expanded. The initial definition you use Hossain et al (2019) contains a number of discrete elements, space, societal challenge and stakeholder involvement. However, it is hard to see the explicit links between all of the case studies and this definition alone, a broader discussion on living labs, encompassing other viewpoints may address this. For example, why is the drone case study simple not a outdoor practical / field work? what make it distinctively a living lab? I think the difference between living labs and other pedagogical approach that take place outside (field work /practical's) is one to emphasis.

The definition of Living Labs are generally similar – with the elements you highlight as the common themes. To better emphasise each of the case studies positioning within the Living Lab we propose linking the individual activities to explicit parts of the Living Lab principles (continuity, openness, realism, empowerment of users, spontaneity). By linking the various activities to elements of the Living Lab this will explain how each activity is situated as part of the Living Lab. Similarly we can link the conceptual framework to each of the case studies (comment 3 below). This could be done by introducing a table marking each case study against the Living Lab Principles, and the framework (plus things like each case studies Intended Learning Outcomes, assessment type etc) used to create them).

 Ultimately these ARE fieldwork/outside practicals; it is the positioning of them within the Living Lab which provides the authentic, student led, congoing activates. "Traditional" outdoor work often has a very specific aim where the exact outcomes and/or observations are "known" by the instructor in advance, and where data collected doesn't "fit" into an ongoing stream of data creation. The difference between work in the Living Lab vs other types of field work and practical work can be emphasised.

2. Case studies - On their own the case studies are all of interest, but they are also all feel quite different narratively. The comment above (comment one) applies here about making sure the 'living lab' elements are specifically identified, for example what is the

societal challenge in the drone and crime scene case studies. The case studies seem to have varying approaches to the amount of detail they include on the approach / activity (such as between case studies 2 and 4), it would probably enhance readability to be a little more consistent here. It is not always clear what the situational context of these case studies are they incurricula / extracurricula? are they assessed formally (noting the emphasis on authentic assessment in the introduction). Finally, there appears to be a very strong emphasis on skill development in several of the case studies, if this is indeed a major purpose for their use. I feel that this not fully explored in the introduction or discussion.

The linking of case studies to Living Lab principles (response to comment 1) will hopefully make the link between each case study and the Living Lab explicit. There will be variety here – the drone example is, for example, a far smaller/more specific exercise than the Environmental Baseline Study. All studies are incurricula (we can add a sentence in the intro to emphasis this, there are some extracurricular aspects to some – e.g. photo contest for the drones), they are linked to different assessment roles (most are formally assessed), this can be made clear in each case study. Very good point on skill development – this needs to be made more explicit as it is one of the major purposes of these learning activities (the learning/use of skills and equipment within a Living Lab are what allows data generation etc. to be spontaneous and realistic and even allows for activities to go 'wrong' whilst skills are still taught!).

3. Framework - The discussion includes a framework for the use living labs. These are framed as a series of questions. I would suggest that this is either not presented as a framework or adapted. Firstly, as this is presumably the authors work some commentary on the pedagogic development of this framework would be useful. Secondly seeing it applied to the case studies in question would also help the reader understand its execution, at present the questions you are ask are not explicit in all the case studies. It is possible these could be introduced before the case studies as the conceptual framework you used in their design.

As indicated above linking the framework to each case study in a table (or similar) would help emphasise how each case study fits into both the Living Lab Principles and into the framework we include. Agree that its introduction before the case studies would help readers understand its execution.

4. Principles - Linked to the above, you introduce the living lab principles in the introduction. As well the direct expression of these in the case studies (see previous comment) how do these principles inform / feed into your case study design (explicitly) and then how to they link to the framework?

The principles can be linked into the framework – moving the framework to come before the case studies would help integrate these two components making it clearer how the principles manifest within the framework.

5. Discussion - The discussion moves to some broader discussions about design and the framework, but I feel there is lots more to be unpicked here. Firstly, the authentic

assessment narrative and learning lab principles from the introduction are not explored fully in the discussion.  There is some links to the individual case studies, but I feel there is more to discuss here, for example case study uses PBL and flipped-classrooms how does the intersection of these pedagogies apply to a living labs context.  I feel more critical evaluation of the living lab examples would be useful for the reader.  Finally beginning line 415 is very Keele specific can this be generalised for a wider readership and, the paragraph beginning line 492 feel disconnected from the living lab discussion that precedes it.

Authentic assessment and the principles can be included in the discussion. Our experience has been that any pedagogies that are applied in labs/practical setting/outdoor learning translate well into the Living Lab – it is the environment, authenticity, ownership, realism (basically the Living Lab Principles) that the Living Lab provides – we can expand on this in the discussion.

Yes – can reword to be less Keele-centric.

Many thanks for taking the time to provide this constructive and thoughtful review.

---

## Author Comment (AC2)

Dear referee,

Many thanks for the time you have taken reading and commenting on this submission. In order to make our response more focused we have provided comment (in red) directly to each point:

Thanks for the opportunity to review this work. The paper presents Living Labs as an important pedagogical tool for higher education learning, outlines strategies for framing activities following this approach, and provide examples of activities carried out in a living lab environment at Keele University since 2006. While the authors introduced the concept of "living labs" thoroughly and provide interesting examples to show how it can be used in education, they do not investigate the concept of living labs or any of the mentioned activities vigorously, and therefore, do not report substantial new results and conclusions. The manuscript, in its current form, reads like a report on "living labs" and not like a scientific investigation of "living labs".

The focus of the submission is the concept of the Living Lab as an educational environment – how we have used it and how others might design similar learning environments. The 'new' results are the framing of education within a living lab (using Living Lab Principles) and the sharing of the framework we used to do this. Living Labs are commonly used to test new ideas and technology, but rarely are curricula embedded within them. Like all education interventions we have used the Living Lab to fit a particular need/purpose (i,e, to provide students with authentic assessments in an environment where they are empowered to learn, is spontaneous, is open etc.) The "evidence" that this has worked are the case studies themselves (our experiences). We hope the submission acts as a dissemination of best practice that will allow other who would like to explore embedding Living Lab Principles into their curriculum can use.

To improve this study and make it publishable in GC, I encourage the authors to consider:

1. Carrying out a qualitative and quantitative assessment of the living lab concept. For example, consider evaluating one or two of the activities already mentioned in the paper for their effectiveness in teaching and learning of specific concepts. Consider comparing them with other forms of "outdoor" activities such as fieldwork or educational fieldtrips.

We appreciate that the submission doesn't contain data commonly found in many other types of papers – however it is relatively common for pedagogic papers exploring concepts and frameworks to be based on author experience, case studies and examples. Indeed, gathering data around many of the Living Lab principles in an educational setting would be wasted effort – e.g. we could ask students if they find gathering and analysing near-surface geophysics data to be more "realistic" or "spontaneous" (for example) than analysing pre-gathered data – but the answer is already plain – and we hope our experiences and dissemination of this would make this clear. Referee 1 suggested more emphasis on skill-based education was acknowledged, which we agree with and believe will help emphasise why education within a living lab is different from other fieldwork (because it is skill based within a real, spontaneous environment that is student led). It would be possible to design such activities outside of the Living Lab too (elements such as continuity and ownership might be a little harder to embed)– this can be emphasised.

2. For each case studies, include the accompanying data, methodology, results, and discussion of results, and consider taking an analytical approach to synthesize the individual case studies into a framework. I also agree with Anonymous Referee #1 that the framework should be applied to the case studies to show readers how to use it.

We agree that the framework (and the Living Lab Principles) need better signposting/integration within each of the case studies (please see the response to ref1). It might be interesting to consider a project comparing student experiences in the Living Lab, at different levels, from different disciplines, with differing amounts of time spent in the Living Lab etc. But we feel this would certainly be another study itself.

I also have a few minor edits and comments, all listed below.

Line 12 – Spell out high education once in the paper (HE)

Line 42 – Please give 1-2 examples (with references) of the innovations that provide platforms for efficient/effective learning environments.

Line 128 – When using terms such as "our student body" and "we", are you referring to a specific group of people or are you using these terms more generally? From how this is written, I take the former to be true. Also, the reference (Ofs, 2020) does not appear in the reference list.

Line 134 – Check grammar: "…are well documented…"

Line 144 – Informal language, consider revision: "…the experiment/test isn't compromised…" – change to "the experiment is not compromised" – same issue in line 537

Line 149 – check grammar: "…but lends itself moreover to effectively transdisciplinary working…"

Table 1 should appear earlier (page 4, for example).

Line 190 – Define FHEQ – not everyone is familiar with this abbreviation. Same with MJCA in line 191. All abbreviations should be defined at least once in the paper.

Line 390 – Not clear why this case study is called "COVID-19 fieldwork" when the actual topic is Climate Change.

The topic of the field course that was switched to run on campus as a response to COVID-19 restrictions focusses on Climate Change.

Line 515 – "Education in these areas…" which areas? Needs clarification.

Line 533 – Give 2-3 examples of the inclusivity/diversity issues mentioned in this sentence.

Line 535 – The sentence needs a verb.

Thank you for these – we can action them all.

Again, many thanks for the time you have taken to read and comment on the submission, we can understand your concerns around the lack of quantifiable data but we would argue that the submission provides the dissemination of a useful (and effective) innovation based on the experience of the authors. The intervention is framed by pedagogic theory.

---

## Author Comment (AC3)

Dear referee,

Many thanks for the time you have taken reading and commenting on this submission. In order to make our response more focused we have provided comment (in red) directly to each point:

Thank you for the opportunity to read your work. For the most part I found this to be a well-presented account of the Living Labs that have been developed at Keele University, which was well structured and engaged with the appropriate literature.

However, at the moment I do not feel that this manuscript is ready for publication in *Geoscience Communication*. The main reason for this is that the work that is presented here is very descriptive. There is little formal reflection, and it is also unclear what the 'success' or impact of the programme of activities has had on both the student and the staff that have been involved in these labs to date. As such, while it is interesting to read about these initiatives, it is unclear how they are advancing the field, and also how (and why) others might adopt such an approach at their own institutions.

We appreciate that the manuscript does not include primary data asking about staff and student perceptions, however we disagree that there is little reflection – the framework these activities are based on was drawn from reflective practice, as were the case study activities themselves (which we agree are quite descriptive in nature). In terms of 'success' or impact, these activities have been evidenced – the impacts/success includes providing students authentic experiences (realism, spontaneity etc.) within the environment in which they live and learn (these are well documented and we would suggest it would be "evidencing the obvious" to pursue data asking students to reflect on these authentic experiences vs "other" pedagogical approaches). The main impact here is embedding these activities into a wider Living Lab – and using the Living Lab as an educational setting (we are aware that this does occur elsewhere – but not on the scale we have attempted to conduct, nor is there currently a framework for individuals to work to (Ref 1 suggested we make this more obvious and link the activities to the framework in a more coherent fasion, which we think would be useful). The "data" we provide is very much conceptual, framework and thematic (e.g. the linking of activities to pedagogic concepts and ideals). This said, we do have access to student, and staff, feedback on the case study/working in the Living Lab etc. This is through Module Evaluations, plus some data gathered from affiliated scholarly projects (with appropriate ethical approval). We have not included this for the reasons stated above, but it would be possible to. Much of this data is somewhat generic and response numbers are limited.

I would strongly encourage the authors to revisit this work and to conduct a detailed study with the students and staff that were involved in these programmes to assess their feedback and evaluate the impact of the Living Labs. The results of these surveys could then be used to contextualise the impact that these Living Labs are having and would also

help to move the findings of the current work from beyond anecdotal evidence to something more formalised. The results from such a survey (or focus group or series of interviews) could also be used to frame the labs and to present a series of recommendations for the development of future activities at both Keele University and beyond.

A detailed perception study would be another manuscript itself – which would require a framework such as this to be available first. The suggestion of recommendations for the development is something that we agree could be included here (from the staff involved in the running of the activities) - student recommendation would be another interesting angle that would probably require a further project phase/output?

I hope that these comments are not too disheartening, as it is really interesting to hear about the work that is being done in these Living Labs. With further reflection, evidence, and framing I believe that this work will be of great value to the wider Geoscience Communication community.

Thank you for these kind words, the review process is always an interesting one! Again, many thanks for the time you have taken to read and comment on the submission.